# Machine Learning Estimation of the Phase at the Fading Points of an OFDR-Based Distributed Sensor

**DOI:** 10.3390/s23010262

**Published:** 2022-12-27

**Authors:** Arman Aitkulov, Leonardo Marcon, Alessandro Chiuso, Luca Palmieri, Andrea Galtarossa

**Affiliations:** 1Department of Information Engineering, University of Padova, Via G. Gradenigo 6/B, 35131 Padova, Italy; 2CERN—European Organization for Nuclear Research, Esplanade des Particules 1, 1211 Meyrin, Switzerland

**Keywords:** DAS, optical fiber, Rayleigh scattering, machine learning, OFDR

## Abstract

The paper reports a machine learning approach for estimating the phase in a distributed acoustic sensor implemented using optical frequency domain reflectometry, with enhanced robustness at the fading points. A neural network configuration was trained using a simulated set of optical signals that were modeled after the Rayleigh scattering pattern of a perturbed fiber. Firstly, the performance of the network was verified using another set of numerically generated scattering profiles to compare the achieved accuracy levels with the standard homodyne detection method. Then, the proposed method was tested on real experimental measurements, which indicated a detection improvement of at least 5.1 dB with respect to the standard approach.

## 1. Introduction

In recent years, research interest in fiber-optic distributed acoustic sensing (DAS) has experienced significant growth in many applications [1], including earthquake detection [2,3], pipeline monitoring [4], railway integrity control [5], gas detection [6], and temperature measurement [7,8]. The advantages of using this method include portability, immunity to electromagnetic interference, and flexibility.

DAS configurations are based on monitoring scattering pattern variations at every position along an optical link. External factors such as acoustic vibrations can induce changes in the refractive index (RI) at different points along the fiber, altering the phase of propagating light [9]. Therefore, the external impact can be quantified by comparing the perturbed scattering profile with the reference state. In order to reliably locate the perturbation point, DAS needs to achieve high spatial resolution.

In optical time-domain reflectometry (OTDR), the fiber is interrogated with a series of pulses, and the time delay between the signal and its reflection, as well as the received power, is used to obtain scattering due to each position along the link [10]. A shorter pulse duration leads to higher spatial resolution. At the same time, the pulse energy is reduced, resulting in a lower signal-to-noise ratio (SNR) [11]; therefore, a compromise between spatial resolution and the SNR is required. By contrast, optical frequency-domain reflectometry (OFDR) interrogates the optical channel with a linearly frequency-swept signal [12]; hence, higher spatial resolution can be achieved by widening the range of the sweep, avoiding the degradation of the SNR.

Several applications have recently been explored by researchers of OFDR-based distributed sensing, including a 25 km long strain sensor with a resolution of 2.5 m [13], a sensor designed for monitoring the thermal conditions of insulation oil [14], a system for measuring the vibrations of oil tanks in transformers [15], and a pressure sensor utilizing a germanium-doped optical fiber core [16]. An OFDR-based DAS system for extracting phase variations induced by acoustic vibrations was reported in Ref. [17], which is the basis for the current work. The advantage of this DAS framework is that the acoustic signal is measured during the frequency scan, enabling high acoustic bandwidth.

Fading is a phenomenon that commonly occurs when working with DAS, regardless of its specific implementation [18,19]; the observed backscattering is affected by interference from reflections caused by the random irregularities of the optical channel [20], leading to the signal deterioration at some positions. Although OFDR-based distributed measurements tend to be intrinsically more robust to fading, this still affects the results. Compared to implementations based on measuring the phase difference, the detection schemes that utilize the cross-correlation technique can mitigate the effects of fading [21] but they also limit the range of detectable frequencies to the scan rate of the source.

Recently, several papers have proposed methods for mitigating the effects of fading in distributed OFDR configurations designed for phase measurements. Fading can be lowered by averaging the scattering patterns across the cores of a multi-core fiber [22]. However, the method is intrinsically incompatible with the DAS framework considered in this paper, since it operates on a single-core fiber. In another approach, the phase signals extracted from the different subsections of a fiber are compared using a sliding window [21]. In the windowed regions, the average of the signals with the highest degree of similarity is used to correct the rest of the signals. Still, as the method is limited by the scan rate, it is not applicable to the considered DAS framework, where the acoustic measurement is performed during the frequency scan.

There is a growing interest in machine learning applications in distributed sensing [23,24]. For instance, vibration event recognition was implemented using relevance vector machines [25], support vector machines (SVMs) [26], and a combination of convolutional neural networks (CNNs) and SVMs [27]. The measurement of corrosion in pipelines with the assistance of CNNs was also shown to be feasible [4]. In Ref. [2], it was demonstrated that three models based on multilayer perceptrons, CNNs, and a combination of CNNs and long short-term memory (LSTM) networks were capable of classifying earthquake signals. It was also shown that the classification of seismic activity can be implemented using a model trained on a synthetic dataset [28]. In the distributed humidity sensor based on the Brillouin spectrum, linear regression was applied to differentiate between humidity and temperature measurements [29].

Although machine learning techniques are increasingly applied to fiber sensing, to the best of our knowledge they have never been used to mitigate the fading problem. In actual fact, a neural network trained with the fiber profiles under several different perturbations can be applied to undo the disruptions in the shape of the detected scattering pattern. However, the majority of the discussed sensing implementations are classification problems. For our purpose, a model capable of managing temporal sequences as input and output is needed. Deep convolutional networks that utilize LSTM put a significant strain on the computation resources during the training [30]. In adjacent optical applications, spatial phase information in the form of image data can be processed with a CNN [31,32]. In other areas, models based on the U-Net architecture have demonstrated promising results in working not only with image data [33] but also with temporal sequences [30,34].

In order to train such a model, a reliable training dataset is required. However, in OFDR, it is difficult to experimentally collect a sufficient amount of diverse scattering profiles subjected to acoustic vibrations because this would require the implementation of many different links using different fibers. Instead, it is more practical to generate a synthetic training set, which is a viable approach sometimes adopted in machine learning. For example, in the generative adversarial network (GAN), the models themselves can be used to generate new samples [35,36], but other configurations can be trained with data simulated by applying theoretical models. Synthetic images [37] and seismic signals modeled with the Ricker wavelet [38] were used to train denoising CNN implementations in the areas of electron microscopy and seismic imaging, respectively. A CNN model for the phase and frequency correction of magnetic resonance spectra was also trained with a simulated set [39]. In the case of the DAS application, another advantage of training the model with simulated scattering patterns instead of real ones is that the neural network is not limited to a specific probing setup or conditions used to obtain the samples. As a result, the network would be able to exhibit more generalized behavior.

From this perspective, this paper reports a machine learning approach for enhancing the extraction of phase variations in OFDR-based DAS systems. Firstly, the theoretical background behind the simulation of the stressed Rayleigh scattering profiles is discussed, which is followed by the descriptions of the standard homodyne and the neural network perturbation detection schemes. Then, the performance of the neural network is verified with a set of simulated scattering patterns. Finally, the new model is also tested on a real set of backscattering profiles, which were measured from fibers subjected to acoustic perturbations under experimental conditions.

## 2. Modeling of Rayleigh Backscattering in Perturbed Fibers

One of the main difficulties in applying machine learning methods is the need for a reliable training set. As in the case at hand, it is often difficult to access an experimental dataset large enough to guarantee reliable training. When this happens, a viable alternative is to use a synthetic training set that is generated by numerically simulating the system. Of course, the quality of the training depends on the quality of the numerical model. For this reason, in this section, we describe the model specifically implemented to simulate OFDR-based DAS measurements.

The Rayleigh scattering profiles of an optical channel subjected to different perturbations are simulated in order to produce the training set for the network. The simulated channel consists of two fiber links of lengths $L_{1}$ and $L_{2}$ connected by a stretcher. As demonstrated later, considering only one perturbation point is not limiting because the data analysis starts by evaluating the perturbation-induced phase delay accumulated up to the fiber section under analysis. Whether this phase delay is due to a single perturbation point or a sequence of them has no impact on this first step of the analysis.

In OFDR, the fiber is interrogated by a tunable laser that is linearly varied in frequency. The probing signal can be defined as $a_{0}\left(t\right)=c\left(t\right)\exp\left(j\left(\pi \sigma t^{2}+2\pi \nu_{0}t\right)\right)$, where c(t) is the baseband component, σ is the frequency sweep rate, and $\nu_{0}$ is the reference carrier frequency. In all of the cases considered in this paper, the measured distance does not exceed the coherence length of the source; hence, the effects of the laser phase noise are neglected in the backscattering model. The signal at the end of the unperturbed section $L_{1}$ is equal to
(1)$$a_{1}\left(t\right)=e^{-j\beta_{0}L_{1}}a_{0}\left(t-\beta_{1}L_{1}\right),$$
where $\beta_{0}$ is the propagation constant and $\beta_{1}$ is the inverse group velocity. As this transmitted signal passes through the subsequent stretcher, the light entering the second link section of the length $L_{2}$ is
(2)$$a_{2}\left(t\right)=e^{j\phi (t)}a_{1}\left(t\right),$$
where ϕ(t) is the phase variation induced by the stretcher.

Generally, the light backscattered by a fiber section of length *L* illuminated by an arbitrary probe light a(t) can be modeled using the function $B_{L}\left[\cdot \right]$ defined as
(3)$$B_{L}\left[a\left(t\right)\right]=\sum_{k}^{}c_{k}e^{-j2\beta_{0}z_{k}}a\left(t-2\beta_{1}z_{k}\right),$$
where $c_{k}$ is the Rayleigh backscattering coefficient of the *k*th scattering center and $z_{k}$ is its position. Thus, the light backscattered by the first fiber section is $b_{0}\left(t\right)=B_{L_{1}}\left[a_{0}\left(t\right)\right]$. Similarly, using Equations (Equation 1) and (Equation 2), the backscattering generated in the second fiber section by $a_{2}\left(t\right)$ is
(4)$$b_{2}\left(t\right)=e^{j\phi (t)}e^{-j\beta_{0}L_{1}}B_{L_{2}}\left[a_{0}\left(t-\beta_{1}L_{1}\right)\right].$$

Then, this signal backpropagates through the stretcher and the first section so that the full reflection due to the second section is
(5)$$b_{1}\left(t\right)\approx e^{j\phi (t)}e^{-j\beta_{0}L_{1}}b_{2}\left(t-\beta_{1}L_{1}\right),$$
where we assume, as usual, that the phase variation in the perturbation is negligible within the round trip time across the second fiber section. Hence, the total backscattering of the stressed fiber can be defined as $b_{str}\left(t\right)=b_{0}\left(t\right)+b_{1}\left(t\right)$.

The total backscattering in the state of rest, $b_{ref}\left(t\right)$, is obtained by setting ϕ(t) equal to zero and then applying the same series of formulas. Despite the degree of randomness contributed by the scattering centers and Rayleigh coefficients, the reference pattern is expected to be constant if any outside impact is avoided. Hence, it can be used as the signature pattern of the fiber [40].

## 3. Estimation of Phase Difference

### 3.1. Standard Detection

For completeness, we briefly review the standard detection technique described in Ref. [17]. Firstly, the signals in the spatial domain are separated into shorter channels so that by considering each channel individually, the perturbed point can be identified. To achieve this, the detected signal $b_{str}\left(t\right)$ is transformed into the frequency domain using a fast Fourier transform (FFT). The resulting signal, $B_{str}\left(f\right)$, is equivalent to the spatial domain representation $B_{str}\left(z\right)$, since the frequency is directly proportional to the distance through $f=2n_{eff}\sigma z/c_{0}$, where $n_{eff}$ is the effective refractive index and $c_{0}$ is the speed of light. A spatial window function W(z) is applied to isolate each channel from the overall signal:(6)$$B_{str,n}\left(z\right)=W_{n}\left(z\right)\cdot B_{str}\left(z\right),$$
where *n* indicates a particular channel that needs to be isolated. For each *n*, the window $W_{n}\left(z\right)$, which is a Tukey window in this case, has cutoff points defined by the positions where the channel $B_{str,n}\left(z\right)$ is supposed to start and end. The length, $L_{W}$, of the region isolated by the window is related to the acoustic bandwidth of the filter, $B_{a}$:(7)$$L_{W}=\frac{B_{a}c_{0}}{2n_{eff}\sigma }.$$

Inverse FFT is applied to each of the channels, obtaining their time-domain form $b_{str,n}\left(t\right)$. The same series of steps is repeated for the reference signal to produce $b_{ref,n}\left(t\right)$. Then, the phase difference at the *n*th channel, $\theta_{n}\left(t\right)$, can be determined as
(8)$$e^{j2\theta_{n}\left(t\right)}=\frac{b_{str,n}\left(t\right)}{b_{ref,n}\left(t\right)}.$$

Ideally, $\theta_{n}\left(t\right)$ would be equal to the applied perturbation, ϕ(t). Given the fiber of length *L*, the overall number of channels is $N=L/L_{W}$. The number of points in each isolated channel is
(9)$$N_{t}=\frac{4L_{W}n_{eff}\sigma T}{c_{0}}=2TB_{a},$$
where *T* is the duration of the frequency sweep.

It has to be noted that the phase in Equation (Equation 8) is the total one accumulated up to the considered channel. Nevertheless, local information can be easily retrieved by assessing the difference
(10)$$\Delta \theta_{n}\left(t\right)=\theta_{n+\Delta n}\left(t\right)-\theta_{n}\left(t\right),$$
where $\Delta n=\frac{L_{W}+\Delta L}{\Delta z}$, ΔL is the separation between the windows that must be lower than *L*, and Δz is the spatial resolution of the OFDR distance axis.

### 3.2. Neural Network Detection

The fading can occur due to the division in Equation (Equation 8). Hence, this step is replaced with computations using a neural network in order to maintain the continuity of the phase evolution along the fiber. The neural network configuration is tested and compared with the standard detection.

One fiber profile is separated into *N* spatial channels; the network operates on each channel independently. A single spatial channel contains two complex traces $b_{str,n}\left(t\right)$ and $b_{ref,n}\left(t\right)$, which correspond to the perturbed state and the reference state. The length of each trace is $N_{t}$ time samples, selected according to Equation (Equation 9) to achieve the required acoustic bandwidth. Hence, the input has a shape of $2\times N_{t}$. The data are further split into real and imaginary components, doubling the first dimension of the shape. Therefore, the overall shape of the complex-valued input is $4\times N_{t}$. The output of the network is the phase difference between the stressed trace and the reference trace; thus, the first dimension of the input transforms from 4 to 1, since the phase difference is a real value. As a result, the output of the neural network is an array of $N_{t}$ values corresponding to the phase variations that occurred in the considered $N_{t}$ time samples.

The data are processed by a series of convolutional down- and upsampling blocks, as shown in Figure 1a. The architecture is similar to the U-Net model reported in Ref. [41]. The first convolutional layer is used to increase the number of input features from 4 to 32. Each downsampling block (as shown in Figure 1b) consists of two convolutional layers with a one-dimensional kernel size of 3. The first layer uses a stride equal to 2 in order to implement downsampling. The architecture is mirrored by each upsampling block, as its first layer utilizes a transposed convolution in order to increase the number of samples. Each convolutional layer in the network uses 32 nodes, except for the last layer, which uses 1 node to obtain a single channel for the final output. Each layer is also followed by the ReLU activation.

Normalization to a range between 0 and 1 is applied to the inputs of the network, $x_{i,normalized}=\left(x_{i}-{\bar{x}}_{-}\right)/\left({\bar{x}}_{+}-{\bar{x}}_{-}\right)$, where $x_{i}$ is a value from a set *x* that needs to be normalized and ${\bar{x}}_{-}$ and ${\bar{x}}_{+}$ are the averages of the negative and positive values in the set, respectively.

#### Network Training

For the training of the network, 160 fiber profiles, each subjected to a different perturbation ϕ(t), are simulated. The overall length of each simulated fiber is 10 m and the length of the isolated spatial channels is 2 cm, corresponding to 500 channels; the other simulation parameters are summarized in Table 1. Perturbation is applied in the middle of the fiber, which means that 250 channels are useful for the model training; therefore, due to the overall number of synthetic fiber profiles, there are 40,000 input traces in the training set.

Given the effective refractive index, sweep rate, and sweep duration shown in Table 1, Equation (Equation 9) stipulates that the length of each trace is equal to 500 points. Substituting $L_{W}$ into Equation (Equation 7), the sampling frequency equal to 2.5 kHz is obtained; thus, as per the Nyquist theorem, the maximum detectable frequency is 1.25 kHz. During the training, perturbation signals ϕ(t) at frequencies in a range from 0 to the detection limit are used. The number of frequency components in different signals is varied from 1 to 4. The fiber length, *L*, determines the number of isolated channels in a single fiber profile. It is directly related to the number of available input traces but has no impact on their size, $N_{t}$. Moreover, after training, due to the constant kernel size of the convolutional layers, the input of the model is not going to be limited by $N_{t}$ or any of the parameters it is dependent on, including $L_{w}$, $n_{eff}$, σ, and *T*, as per Equation (Equation 9). Hence, the network performance is scaled with $L_{w}$ and the corresponding frequency limit, and it is possible to apply the model to traces of varying lengths, regardless of the training parameters listed in Table 1.

Since the exact value and application point of the generated stress are available, the nominal phase $\theta_{n}\left(t\right)$ is known for every point of all channels *n* and is used as the ground truth for the network training. The smoothness of the targeted output is supposed to help the network learn how to maintain the continuity of the detected phase.

## 4. Results

### 4.1. Numerical Verification

Another set of 50 stressed fiber profiles, which resulted in 12,500 channel traces, was simulated to compare the standard method and the tested neural model. The frequencies and the amplitudes of the simulated perturbations were evenly distributed in ranges between 0–1200 Hz and 0–1 rad, respectively. For each channel of every signal, the mean error between the detected phase difference and the corresponding nominal waveform, *E*, was calculated to characterize the impact of noise on the performances of the tested detection methods. Then, the values ε that comprised the resulting error sets were used to estimate the cumulative distribution function (CDF) equal to F(E)=P(E≤ε). Figure 2a presents the survival function P(E>ε)=1−F(E), i.e., the probability that the mean error was larger than a value ε. In general, the neural network performed better, indicating a 6 dB improvement over the standard method. Figure 2b uses the same error metric to show the dependency of the accuracy on the perturbation frequency. The performances of the methods degraded as the perturbation approached the detection limit, but the network consistently resulted in error values lower than those of the standard approach.

In principle, the performance of a neural network depends on its complexity and the quality and size of the training dataset. In our numerical tests, we also verified that, for the complexity of the selected neural network, the performance had not significantly improved when utilizing datasets larger than the one used for the current model.

### 4.2. Experimental Tests

The performance of the neural network was also tested on the Rayleigh scattering profiles measured with a commercial OBR (Luna OBR 4600, Luna Innovations). Due to the misalignment caused by a linearizing interferometer used in the OBR, the time axis of the applied perturbation needed to be corrected using the compensation technique from Ref. [17] before the analysis. According to the method, another reference signal is applied to the fiber concurrently with the perturbation. This known signal is extracted and the difference between the measured time axis and the actual time axis of the reference is estimated. Then, this value is used to resample the time axis of the whole OBR measurement.

First, a sinusoidal vibration at 41 kHz was applied at a distance of 8 m along a 12 m long fiber using a loudspeaker. The perturbations obtained using the compared methods with a window size of 75 cm and a frequency limit of 45 kHz can be seen in Figure 3. The window separation, ΔL, was 15 cm. The reference frequency of the signal used to correct the OBR time axis was 3 kHz.

Upon visual inspection along the vertical axis in Figure 3a, the signal was heavily deteriorated by noise around the perturbation frequency. However, the neural network was able to lower the intensity of this noise pattern, making the perturbation frequency significantly more prominent compared to the standard method. The efficiency of the proposed method can be evaluated by measuring the mean value of the SNRs calculated for each channel of the perturbed region. With respect to the standard method, the neural network achieved an SNR improvement equal to 7.3 dB. An example of the detected spectrum, collected at the 8 m position, can be seen in Figure 4, which also corroborates the decrease in noise and shows that the perturbation frequency was more pronounced.

In the second test, a more complex perturbation with several frequency components, as shown in Figure 5, was applied at a distance of 15 m along a 50 m long fiber. The perturbation is defined as $A_{1}\sin\left(2\pi f_{1}t\right)+A_{2}\sin\left(2\pi f_{2}t\right)\cdot \exp\left(-0.5{\left(t-\mu \right)}^{2}/\Lambda^{2}\right)$, where $A_{1}=27.69\times 10^{-2}$, $A_{2}=18.46\times 10^{-5}$, $f_{1}=150$ kHz, $f_{2}=120$ kHz, μ=0.5 ms and Λ=0.4 ms. The detection was conducted using a window size equal to 4 m, which yielded a frequency limit of 250 kHz. The window separation, ΔL, was 25 cm. The reference frequency for correcting the OBR misalignment was 150 kHz, which was within the detection limit. Figure 6 presents the 2D maps where the obtained frequency spectra are plotted for each spatial channel; to enhance the visibility of the result, only subbands of 100 Hz around the harmonic frequencies are shown. The proposed method successfully detected the most prominent frequency components shown in Figure 5b.

In the perturbed section, the average SNR of the neural network was 5.1 dB higher than that of the standard approach. The improvement achieved by the tested model in this experiment was lower than in the case of the 41 kHz perturbation shown in Figure 3b, which can be explained by the higher complexity of the detected signal. The observations about noise reduction can be supported by the examples of the detected perturbations shown in Figure 7, which were collected at a distance of 20 m. The neural network was also able to provide more efficient detection of the perturbation components at frequencies lower than 117 kHz.

## 5. Conclusions

The new machine learning method for extracting the phase difference from the Rayleigh scattering profiles measured by OFDR in perturbed fibers has been proposed and verified with a set of real fiber profiles subjected to acoustic vibrations during the experiments.

Firstly, a specific mathematical model that describes the scattering by OFDR signals in stressed fibers has been presented and used to generate synthetic data comprising scattering patterns under perturbations of different frequencies. These simulated profiles were applied to train a neural network, which is based on the U-Net architecture and receives real and imaginary components of the Rayleigh scattering as the input. After validating the performance using another set of synthetic scattering profiles, the network was also tested on real experimental measurements of scattering in perturbed fibers, achieving at least a 5.1 dB improvement over the standard method.

## Figures and Tables

**Figure 1 sensors-23-00262-f001:**
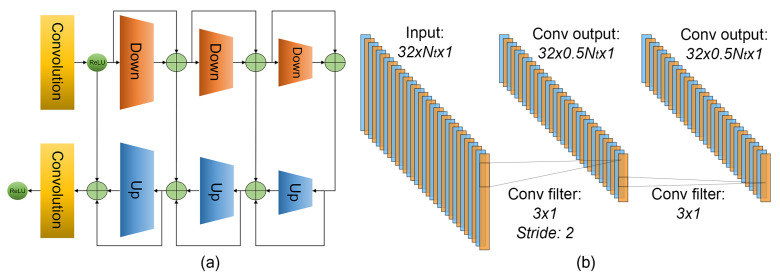
(**a**) Network layers of convolutional down/upsampling configuration. (**b**) Architecture of single downsampling block.

**Figure 2 sensors-23-00262-f002:**
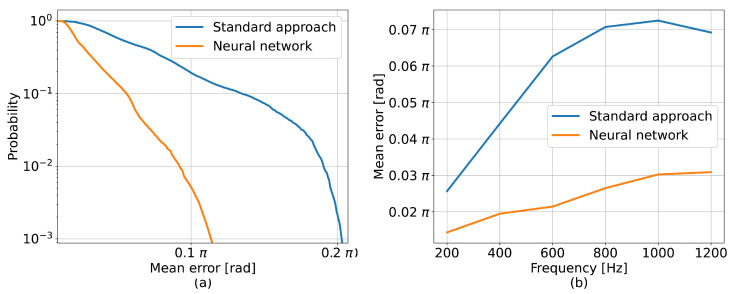
(**a**) Survival function for mean errors between the detected phases and their best fits. (**b**) Relationship between mean best fit error and frequency.

**Figure 3 sensors-23-00262-f003:**
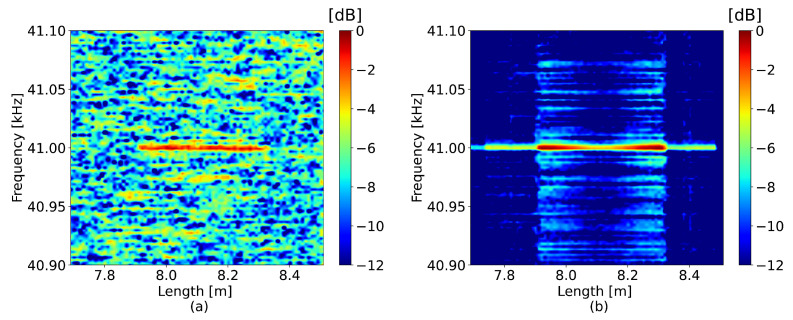
41 kHz perturbation detected by (**a**) the standard method and (**b**) the neural network.

**Figure 4 sensors-23-00262-f004:**
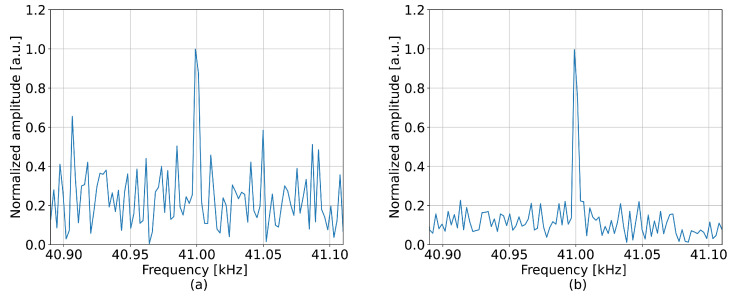
Frequency spectrum detected when measuring 41 kHz perturbation at 8 m by (**a**) the standard approach and (**b**) the neural network.

**Figure 5 sensors-23-00262-f005:**
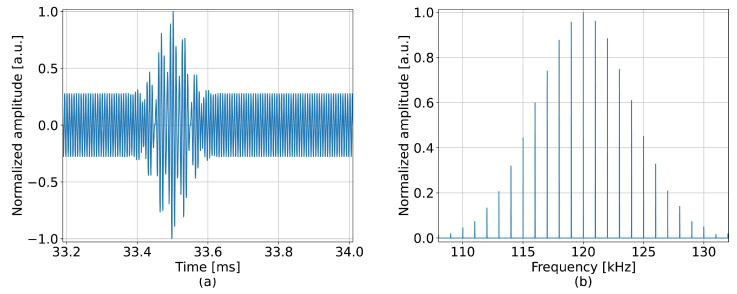
Perturbation of increased complexity applied experimentally (**a**) in the time domain; (**b**) in the frequency domain.

**Figure 6 sensors-23-00262-f006:**
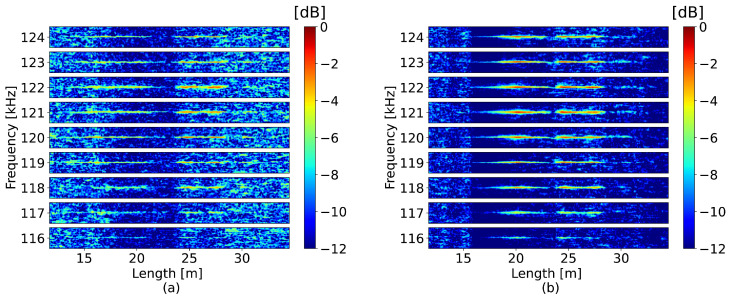
Perturbations of increased complexity detected by (**a**) the standard method and (**b**) the neural network (demonstrated spectral width around each component was 100 Hz).

**Figure 7 sensors-23-00262-f007:**
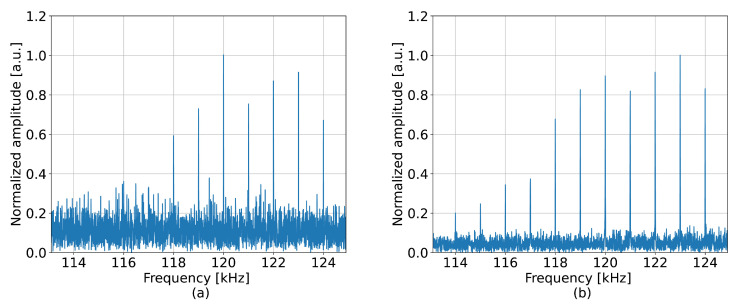
Frequency spectrum detected when measuring perturbations of increased complexity at 20 m by (**a**) the standard approach and (**b**) the neural network.

**Table 1 sensors-23-00262-t001:** Simulation parameters.

*L* (m)	$L_{W}$ (cm)	$n_{\mathrm{eff}}$	σ (THz/s)	*T* (s)
10	2	1.5	12.5	0.1

## Data Availability

The data presented in this study are not publicly available at this time, but may be obtained upon reasonable request from the authors.

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
