# Peer review of "Machine Learning Estimation of the Phase at the Fading Points of an OFDR-Based Distributed Sensor"

_sensors, 2022, doi:10.3390/s23010262_

Round 1

Reviewer 1 Report

The paper proposes and demonstrates a machine learning method for phase extraction in distributed acoustic sensors based on the phi-OFDR. The topic is of interest, and the results are encouraging. However, some parts should be improved for clarity before publication. The major issue, in my opinion, lies into the operation of the neural network. According to Section 3.2 (neural network detection), the neural network (NN) receives the reference trace and the stressed trace, corresponding to one channel and one time instant. As one channel has Nt points, and considering the complex nature of the input signals, we get a 4 X Nt size for the NN input. Therefore, there is no reference about the time dependance of the perturbation. Later, in section 3.2.1, the authors refer to a time-dependant perturbation for the NN training, as il the NN had memory of the past values of the perturbation. Similarly, the statement in line 188, according to which the improvement achieved by the NN is lower in case of higher complex signals, seems to suggest that the NN is somewhat built to process the temporal profile of the perturbation, rather than the reference and stressed traces at a specific instant. If the NN operates by recovering the phase of each temporal snapshot of the signal, why the temporal complexity (i.e., the frequency content) of the signal should affect the reconstruction quality?

 There are also other points that must be clarified:

1)      Line 39. The authors state that all DAS configurations are subjected to fading. As a matter of fact, conventional OFDR configurations based on cross-correlation are less subject to fading, as reported in W. Feng, M. Wang, H. Jia, K. Xie and G. Tu, "High Precision Phase-OFDR Scheme Based on Fading Noise Suppression," in Journal of Lightwave Technology, vol. 40, no. 3, pp. 900-908, 1 Feb.1, 2022, doi: 10.1109/JLT.2022.3142164. Please comment on this.  

2)      Line 72. The quantity bo(t) is not defined. Maybe the correct equation is bstr(t) = b1(t)+b2(t).

3)      In page 3, the authors refer for the first time to the “sweeping rate”. It should be clarified that the OFDR measurement is carried out by linear sweeping the frequency of a tunable laser.

4)      Line 96. The NN is built to produce a phase difference per each sampled point along the fiber. Therefore, after NN processing we obtain Nt points per channel. Wouldn't it be more efficient to build the NN in order to get a single value of the phase difference per channel?

5)      The authors should comment if their method could be improved in some way, for example by increasing the dataset used for training.

6)      The authors should compare their method with other ones recently proposed for fading suppression in phi-OFDR, such as  “High Precision Phase-OFDR Scheme Based on Fading Noise Suppression," JLT 2022, or Y. Feng et al., "Multicore Fiber Enabled Fading Suppression in φ-OFDR Based High Resolution Quantitative DVS," in IEEE Photonics Technology Letters, 2022.

Reviewer 2 Report

The paper ‘Machine Learning Prediction of Phase at Fading Points of OFDR-based Distributed Sensor’ reports a machine learning approach to estimate the phase at fading points, which may help improve the detection. The novelty of this manuscript is not clear and the following comments may help to improve it:

1.       What is the advantage of Machine learning compared to traditional signal processes regarding your dataset? From Fig. 4 & 6, it is not feasible to use the trained ML model based on simulation process and data.  This model may not be the best model for experiment data; how about other neural networks and models?  It looks like the experiment data training is not complicated and will be more convincing.

2.       Page 2, Line 42, Machine learning in the distributed sensor has been reported by multiple researchers, but literature reviews are lacking in the introduction.

3.       OFDR is based on the continuous wave; how to deal with the fading if multiple perturbations occur in case the neural network may be invalid?

4.       What is the coherent length of the source?

5.       Eq 3., if the ck is the Rayleigh coefficient, what does the power of the exponential ‘-j2β0zk’ mean?

Round 2

Reviewer 1 Report

The authors have clarified the issues raised in my previous review, which mostly resulted from a wrong interpretation of the algorithm description. 

I reccommend publication of the paper in its present form. 

Reviewer 2 Report

It still needs to be determined why the authors selected the architecture U-Net model and trained with simulated datasets. There are two suggestions:

1) U-Net is a well-designed model and has been used in this manuscript. However, it is not comparable with LeNet, or CNN +LSTM, given the authors' response. Even though the authors stated that "none of them yield results accurate enough," all experiment data and evaluation were missing. How were the networks U-Net, LeNet, and CNN +LSTM built and trained? A clear understanding of the features of neural networks and their corresponding applications are highly recommended. Additionally, there are extensive studies related to U-Net ++ and other U-net – based extensions in the literature. The authors failed to check the literature sightly and discuss the neural networks and trained models scientifically. Where were the training results and parameters of the trained model (not table 1)?

Or 2) If this study is to use one well-trained model, please train the NN model with experimental data and demonstrate the results. Line 83 mentions, 'it is often difficult to access an experimental data set large enough to guarantee reliable training.' On the other hand, 160 fiber profiles are simulated for network training. I did not see the difficulty training with experimental data since the location and frequency components could be simply changed. Please also check the title; the prediction part is missing in the manuscript.

Round 3

Reviewer 2 Report

n/a